

# Ultrasonographic evaluation of intrinsic foot muscle thickness in soccer and basketball players: an observational study

Lorena Canosa-Carro[1], Jorge Hugo Villafañe[1], Unai Torres-Berra[1], Jaime Almazán-Polo[2], Helios Pareja-Galeano[3], Sergio Vázquez-González[1] and Carlos Romero-Morales[1]

[1] Physiotherapy, Universidad Europea de Madrid, Madrid, Comunidad de Madrid, Spain
[2] Faculty of Nursing, Physiotherapy and Podiatry, Department of Physiotherapy, Complutense University of Madrid, Madrid, Spain
[3] Physical Education, Sport and Human Movement, Universidad Autónoma de Madrid, Madrid, Spain

Corresponding author
Jaime Almazán-Polo,
jaalmaza@ucm.es,
jaime.almazanpolo@gmail.com

## ABSTRACT

**Objective**. This study aimed to assess and compare the thickness of specific intrinsic foot muscles (IFM)—abductor hallucis brevis (ABH), flexor hallucis brevis (FHB), flexor digitorum brevis (FDB), quadratus plantaris (QP)—and the plantar fascia (PF) in amateur soccer and basketball players using ultrasonography.

**Methods**. A cross-sectional design was employed, involving 35 male amateur athletes, including 17 soccer players and 18 basketball players. Ultrasonographic imaging was performed to measure the thickness of the IFM and PF in a relaxed position for all participants.

**Results**. Basketball players demonstrated significantly greater thickness in the ABH and FHB compared to soccer players. However, no substantial differences were observed in the thickness of the FDB, QP, or PF between the two groups.

**Conclusions**. The study identified sport-specific differences in the thickness of certain intrinsic foot muscles between soccer and basketball players. These variations may be attributed to the unique movement patterns and biomechanical demands of each sport, highlighting the importance of targeted training and injury prevention programs tailored to the needs of these athletes.

# INTRODUCTION

The importance of sport-specific training in enhancing athletic performance and reducing injury risk is well-documented in sports science literature (*Wang et al., 2023*). Different sports impose varied physical demands on athletes, leading to various adaptations in muscle strength, endurance, and morphology, as well specific approaches to training and rehabilitation programs (*Taylor et al., 2017*; *Nuñez et al., 2021*). Among them, basketball and soccer—two of the most popular sports worldwide—illustrate how varying physical,

technical, and tactical demands shape the athlete's profile depending on the style of play. Basketball is defined as a high-intensity intermittent discipline with frequent jumps, pivots, sprints and changes of direction. These repetitive, high-impact activities lead to specific muscle adaptations, improving explosive power and anaerobic capacity (*McGill, Andersen & Horne, 2012*). In contrast, soccer focuses on endurance, agility, and running skills, requiring both sustained aerobic performance and short bursts of high-intensity activity (*Bortnik et al., 2024*).

Beyond general physiological demands, the biomechanical context of each sport—particularly the interaction between playing surface and footwear—modulates the mechanical loading experienced by foot structures such as the plantar fascia, intrinsic foot muscles, and the medial longitudinal arch (*Siegel, Sproll & Zech, 2025*). These factors have been shown to influence foot stiffness, strength, and morphology, contributing to sport-specific adaptations in intrinsic foot muscle (IFM) characteristics (*Villwock et al., 2009*; *Holowka, Wallace & Lieberman, 2018*). Moreover, basketball involves a greater number of movements in the frontal plane and repeated vertical loading due to jumping demands, whereas soccer players perform more actions in the sagittal plane, such as linear running and multidirectional sprinting (*Domínguez-Díez et al., 2021*). These distinct movement patterns, combined with differences in surface and footwear, affect the direction, magnitude, and frequency of mechanical stress applied to foot structures, potentially leading to divergent musculoskeletal adaptations in athletes from different disciplines (*Domínguez-Díez et al., 2021*). Despite the well-established differences in overall physical demands, studies directly comparing specific muscular adaptations between basketball and soccer athletes remain limited.

Although the role of the intrinsic foot muscles (IFM) in maintaining foot posture and stability is well established—supporting the medial longitudinal arch, modulating load distribution, and coordinating with extrinsic musculature throughout gait—less attention has been given to their clinical assessment and targeted intervention (*Tourillon, Gojanovic & Fourchet, 2019*; *Jastifer, 2023*). Recent evidence suggests that IFM training can improve foot biomechanics, as reflected in reductions in navicular drop ($p = 0.02$) and foot posture index ($p = 0.0003$), along with enhanced postural control (*Wei et al., 2022*). The impact of sport-specific demands on IFM morphology and structural adaptations has not been extensively explored (*Wei et al., 2022*; *Jaffri et al., 2023*). This is particularly relevant in disciplines such as soccer and basketball, where IFM strength and morphology are crucial due to their role in kicking, jumping, and sprinting mechanics (*Girard et al., 2019*). The role of the ankle-foot joint complex in providing stability and generating propulsion during activities that involve accelerations and decelerations requires, among other factors, optimal IFM function (*Jastifer, 2023*; *Jaffri et al., 2023*). Moreover, the plantar fascia is essential for the distribution of forces in foot biomechanics, warranting investigation in terms of its morphology, implications for functional foot behavior, and relevance in sports activities such as sprinting (*Natali, Pavan & Stecco, 2010*). In this sense, controlling deformation of the foot arches depends on the well-coordinated synergy between dynamic active structures, such as the IFM, and passive tensile elements like the plantar fascia (*Kelly et al., 2014*; *McKeon et al., 2015*).

Ultrasound imaging of the morphology of the IFM has shown to be a reliable option compared to the gold standard test, Magenitic Resonance Imaging (MRI), even when comparing reliability between examiners of varying experience, and different positions (*Swanson et al., 2022*; *Franettovich Smith et al., 2019*). Among the advantages of ultrasound, it stands out that it is a non-invasive system, with easier access, portable, which allows these muscle groups of the foot to be effectively examined more readily (*Crofts et al., 2014*; *Franettovich Smith et al., 2019*). Ultrasound imaging has demonstrated good to excellent reliability (intraclass correlation coeficiente (ICC) = 0.76–0.98) for assessing IFM thickness and cross-sectional area (CSA), with low measurement error (standard measurement error (SEM): 0.05–0.09 cm; minimal detectable change (MDC): 0.14–0.24 cm), supporting its use in morphofunctional assessment. Additionally, previous research has reported intra- and inter-rater ICCs consistently above 0.81, with SEMs ranging from 0.05 to 6.5% and MDCs from 0.14 cm to 22.6%, confirming its robustness across evaluators and testing conditions (*Franettovich Smith et al., 2019*; *Fraser, Mangum & Hertel, 2018*).

Thus, the aim of the present study was to evaluate the thickness differences in the IFM—abductor hallucis (ABH), flexor hallucis brevis (FHB), flexor digitorum brevis (FDB), quadratus plantaris (QP)—and the plantar fascia (PF) at its insertion (PF1), midfoot (PF2) and forefoot (PF3) between amateur soccer and basketball players using ultrasound imaging, addressing muscular differences between sports disciplines. Accordingly, the present study hypothesizes the existence of sport-specific morphological adaptations in the IFM and PF, whereby basketball players may exhibit greater thickness in structures associated with repetitive jumping and pivoting, such as the ABH and PF, while soccer players may show increased development of propulsion-related muscles, particularly the flexor hallucis brevis FHB.

## METHODS

### Design

A cross-sectional study was conducted between April and May 2024 to compare the thickness of intrinsic foot muscles and the plantar fascia using ultrasound imaging in amateur soccer and basketball players, with all assessments performed at the end of the regular season. The study adhered to the Strengthening the Reporting of Observational Studies in Epidemiology (STROBE) guidelines for cross-sectional studies to ensure methodological rigor and enhance the reproducibility of the findings (*Von Elm et al., 2008*).

### Sample size calculation

The sample size was estimated a priori using G*Power (v.3.1.9.7), assuming a two-tailed *t*-test for point-biserial correlation, an alpha level of 0.05, a desired power (1–β) of 0.80, and a medium effect size (Cohen's $d = 0.45$), which yielded a minimum required total of 33 participants, achieving an actual power of 0.8007. However, due to logistical limitations and the structure of the study (based on the incorporation of two sports rosters from an amateur basketball and soccer team) the final sample included 35 athletes. This figure was

determined by convenience and includes a 10% margin to account for potential attrition or missing data. Although the sample size did not meet the theoretical requirement, the use of intact team units and the expectation of a medium effect size support the feasibility and interpretability of the findings.

## Participants

This study involved 35 male amateur athletes, comprising 17 soccer players and 18 basketball players. Eligibility criteria included: (1) healthy amateur athletes; (2) active participation in a team with a structured schedule encompassing organized weekly training and competitive sessions; (3) high commitment levels despite the absence of financial compensation; and (4) no history of musculoskeletal injuries to the lower extremity or ankle-foot complex in the past 3 months that could affect sports performance (*Mcauley, Baker & Kelly, 2022*). Players were excluded if they failed to meet all eligibility criteria and possessed a history of lower-limb injuries, surgeries, or conditions that could impact muscle morphology.

## Ethical considerations

The Research Ethics Committee of Universidad Europea de Madrid granted approval for this study (internal code CI [2024–797]). All participants received comprehensive information regarding the study's objectives, procedures, and their right to withdraw without repercussions. Informed consent was obtained in writing, adhering to the principles outlined in the Declaration of Helsinki (*Shrestha & Dunn, 2020*).

## Descriptive variables

Following enrollment in the study, all participants underwent anthropometric assessment, including height and body mass measurement using a calibrated scale (Baxtran RGT, Baxtran, Spain). Participants also self-reported their shoe size (European sizing system) and the average number of hours spent training per week, excluding competition time (typically held on weekends).

## Sonographic evaluation

Ultrasonographic assessments were conducted using a SonoScape E2 ultrasound system (SonoScape, ES) equipped with a linear transducer operating within an 8–13 MHz frequency range and featuring a 55-mm footprint in B-Mode. A clinician with substantial experience (U.T.) (more than 3 years) performed the ultrasound imaging to evaluate the thickness of intrinsic foot muscles (IFM), following standardized protocols to ensure precision and consistency, with muscles assessed in a resting state, given that all measurements were obtained under non-contractile conditions. Ultrasound-based assessment has shown excellent reliability for evaluating IFM architecture across different evaluators and testing conditions. Franettovich Smith M.M. et al. (2019) reported high within-session intra-rater ICC > 0.94 (SEM < 3.6%; MDC < 10.0%) alongside reasonable between-session intra-rater ICC > 0.81 (SEMs < 6.5%; MDCs < 17.9%) (*Franettovich Smith et al., 2019*). For the assessment of the ABH, all participants placed the foot on the examination table in a supine position, maintaining heel contact to control ankle movement. In contrast, for the evaluation of the PF, QP and the FHB, the foot was positioned off the edge of the table,

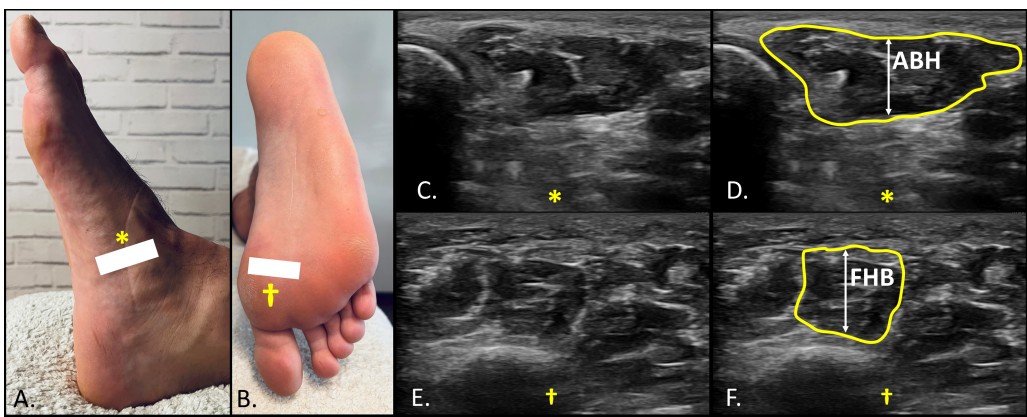

**Figure 1** **Ultrasound imaging of the abductor hallucis (ABH) and flexor hallucis brevis (FHB).** Medial (A) and plantar (B) view of the foot showing the landmarks the reference points for probe location and detection of ABH (A, *) and FHB (B, †); (C) and (D) short axis view for the thickness determination of the ABH distally to the navicular process (*); (E) and (F) short axis view for the thickness determination of the FHB between the superficial location of the flexor hallucis longus tendon and the cortical bone of the first metatarsal (†).

maintaining a 90° angle of dorsiflexion at the ankle, avoiding toe extension and increased tension on the flexor hallucis brevis and the plantar fascia. The IFM were imaged in two areas: the medial side for the ABH in short axis view (Fig. 1A) and the plantar side for the FHB in short axis (Fig. 1B), and the FDB, QP, PF in long axis (Fig. 2A). Muscle thickness was measured as the distance between superficial and deep borders. For ABH, the probe was placed over the medial foot aspect, perpendicular to the long axis, with the ABH visible under the navicular tuberosity (Figs. 1C and 1D). FHB thickness was measured with the probe referencing the first metatarsal bone and the flexor hallucis longus tendon, confirmed by passive flexion-extension of the interphalangeal joint (Figs. 1E and 1F). For PF assessment, participants were positioned prone with the ankle at 90°. The probe was placed longitudinally on the sole to identify the PF, FDB, and QP. PF thickness was measured at the calcaneus (PF1) (Figs. 2B and 2C), midfoot (PF2) (Figs. 2D and 2E), and forefoot (PF3) (Figs. 2F and 2G) by positioning the probe longitudinally at these locations and applying tension to confirm the integrity of these structures during foot dorsiflexion (*Crofts et al., 2014*; *Zaottini et al., 2023*). All ultrasound images were stored in each participant's clinical record and subsequently exported in DICOM format for offline analysis using ImageJ-Fiji software (*Schindelin et al., 2012*). ImageJ is a free, open-source image processing program capable of displaying, editing, analyzing, processing, saving, and printing a wide variety of image types. All images were analyzed by the same evaluator, a physical therapist with experience in musculoskeletal ultrasound and proficient use of the software, who was blinded to group allocation. Participants were assigned coded identifiers during the imaging procedures to ensure blinding throughout the analysis process.
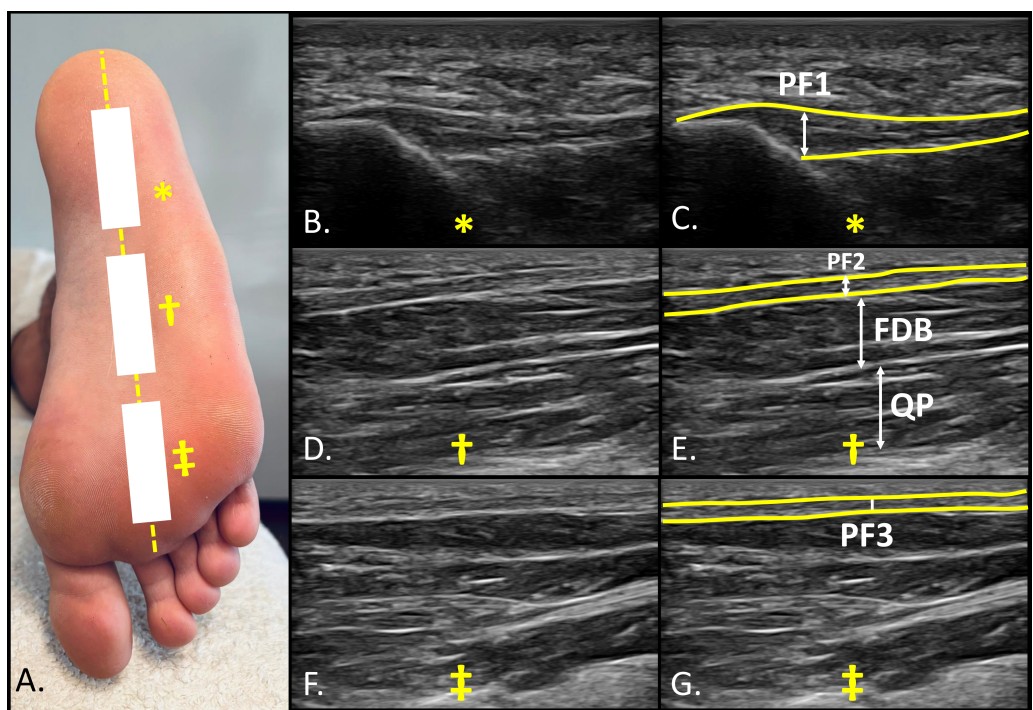

**Figure 2   Ultrasound imaging of the sole of the foot at three reference points.** (A) Plantar view of the sole of the foot as well as landmarks showing the three reference points (*, †, ‡) for probe location and imaging assessment; (B) and (C) plantar fasciae (PF1) at proximal calcaneal insertion (*); (D) and (E) plantar fasciae (PF2), flexor digitorum brevis (FDB) and quadratus plantaris (QP) at the middle third of the foot (midfoot) (†); (F) and (G) plantar fasciae (PF3) at distal portion of the foot (forefoot) (‡).

## Statistics

Data analysis was executed using RStudio (version 1.4, RStudio, PBC, Boston, USA) and Jamovi (version 1.6, The Jamovi Project). The Shapiro–Wilk test assessed data normality ($p > 0.05$). Descriptive statistics were presented as means and standard deviations (SD) for parametric data, and medians with interquartile ranges (IR) for non-parametric data. Group comparisons for parametric data employed the independent samples Student's $t$-test, while the Mann–Whitney U test was utilized for non-parametric data. Levene's test verified the homogeneity of variances. Effect sizes (ES) were calculated using Cohen's d and interpreted as small ($\leq 0.20$), medium (0.21–0.79), and large ($\geq 0.80$), in accordance with conventional guidelines. Subsequently, two multiple linear regression models were performed to examine the association between individual characteristics and the thickness of ABH and FDB. The predictors included in both models were group (basketball *vs.* soccer), age, and height, as these variables showed significant correlations with the dependent variables that demonstrated between-group differences. Correlations were assessed using Pearson's correlation coefficient for normally distributed variables and Spearman's rho for non-normally distributed variables. Prior to analysis, all assumptions for linear regression were verified. Linearity, homoscedasticity, and normality of residuals were confirmed

**Table 1  Sociodemographic data of the sample.**

| Measurement | Soccer ($n = 17$) Mean ± SD | Basketball ($n = 18$) Mean ± SD | *P*-value |
|---|---|---|---|
| Age, y | 23.1 ± 3.53 | 23.6 ± 2.45 | 0.136 |
| Weight, kg | 81.8 ± 8.44 | 78.2 ± 8.0 | 0.204 |
| Height, cm | 1.81 ± 0.06 | 1.86 ± 0.08 | 0.043 |
| BMI, kg/cm$^2$ | 25.1 ± 2.42 | 22.6 ± 1.64 | 0.001 |
| Shoe size | 43.6 ± 1.92 | 45.0 ± 1.50 | 0.018 |
| Week training hours | 4.91 ± 1.68 | 6.83 ± 2.33 | 0.009 |

through visual inspection of scatterplots and histograms, as well as normal probability (P–P) plots. The independence of residuals was assessed using the Durbin–Watson statistic, with acceptable values obtained for both models (DW = 2.19 for ABH and DW = 2.17 for FDB). Multicollinearity was ruled out by checking the variance inflation factor (VIF) and tolerance, with all VIF values below 1.25. For each model, the coefficient of determination (adjusted $R^2$) was reported along with the unstandardized regression coefficients (B), standardized coefficients (β), and significance values ($p$). Statistical significance was set at $p < 0.05$.

# RESULTS

Regarding descriptive variables, significant differences were found in height ($p = 0.043$), body mass index (BMI) ($p = 0.001$), shoe size ($p = 0.018$), and weekly training hours ($p = 0.009$) (Table 1). Significant differences were also observed in the ABH and FDB, with basketball players showing greater muscle thickness in ABH (12.8 ± 0.42 mm; mean ± standard error of the mean) compared to soccer players (10.8 ± 0.39 mm; $p = 0.002$, ES = 1.15). Similarly, FDB thickness was significantly higher in basketball players (11.0 ± 0.17 mm) than in soccer players (8.72 ± 0.40 mm; $p = 0.001$, ES = 1.89). No significant differences were found in the QP or in the plantar fascia measurements at the PF1, PF2, or PF3 between groups (Table 2).

The selection of variables for the two multivariate linear regression analyses was based on preliminary Pearson correlation tests, which revealed significant associations between group and ABH ($p = 0.003$), group and FDB ($p < 0.001$), age and ABH ($p = 0.042$), and height and FDB ($p = 0.031$). Although age was not significantly associated with FDB, it was retained in the model to ensure consistency across analyses. Table 3 shows the results obtained for both multivariate linear regression models. In the case of ABH, the model was statistically significant ($p = 0.005$), with an adjusted $R^2$ of 0.27 and a standard error of 1.66. Among the predictors, only group showed a significant association with ABH thickness (B = 1.56, β = 0.407, $p = 0.015$), while age ($p = 0.233$) and height ($p = 0.337$) were not significant. Similarly. the model for FD Balso reached statistical significance ($p < 0.001$), explaining 45.5% of the variance (adjusted $R^2 = 0.455$), with a standard error of 1.29. Again, group was the only significant predictor (B = 2.23, β = 0.649, $p < 0.001$), whereas age ($p = 0.846$) and height ($p = 0.385$) did not contribute significantly to the model (Table 3).

**Table 2  Ultrasound imaging measurements of intrinsic foot muscles.**

| Measurement (mm) | Soccer (*n* = 17) | Basketball (*n* = 18) | *P*-value (effect size) |
|---|---|---|---|
| | Mean ± SD (95% CI) | Mean ± SD (95% CI) | |
| ABH | 10.8 ± 1.40 (7.52–14.0)[*] | 12.8 ± 1.77 (9.45–15.4)[*] | **0.002 (1.15)**[**] |
| FHB | 9.76 ± 1.40 (7.68–12.8)[*] | 10.1 ± 0.7 (8.9–13.3)[†] | 0.176 (0.47)[‡] |
| FDB | 8.72 ± 1.66 (6.96–13.9)[*] | 11.0 ± 0.29 (10.1–13.1)[†] | **0.001 (1.89)**[‡] |
| QP | 8.63 ± 1.76 (5.14–12.0)[*] | 8.71 ± 0.81 (7.4–10.2)[*] | 0.872 (0.05)[**] |
| PF1 | 1.49 ± 0.29 (0.73–3.33)[†] | 1.49 ± 0.23 (0.9–1.8)[*] | 0.198 (0.44)[‡] |
| PF2 | 1.44 ± 0.25 (0.9–2.1)[*] | 1.42 ± 0.25 (1.0–1.8)[*] | 0.797 (0.08)[**] |
| PF3 | 1.34 ± 0.19 (0.9–1.7)[*] | 1.33 ± 0.05 (1.2–1.4)[*] | 0.909 (0.03)[**] |

Notes.

Abbreviations: ABH, abductor hallucis brevis; FDB, flexor d longus; FDB, flexor hallucis brevis; QP, quadratus plantae; PF1, plantar fascia at calcaneus insertion; PF2, plantar fascia midfoot; PF3, plantar fascia forefoot.

For all analyses, $p < .05$ (for a confidence interval of 95%) was considered statistically significant (bold).

[*] Mean (SD) was applied.

[**] Student's *t*-test for independent samples was performed.

[†] Median (IR) was used.

[‡] Mann–Whitney *U* test was utilized.

**Table 3  Multivariate predictive analysis for ABH and FDB significative values.**

| | Variable | B (Unstandardized) | B (Standardized) | *P* value | VIF | Adjusted $R^2$ | Std. Error |
|---|---|---|---|---|---|---|---|
| ABH | Constant | 4.996 | – | 0.545 | – | 0.270 | 1.66 |
| | Group | 1.559 | 0.407 | 0.015 | 1.161 | | |
| | Age | −1.12 | −0.193 | 0.233 | 1.172 | | |
| | Height | 3.916 | 0.159 | 0.337 | 1.241 | | |
| FDB | Constant | 1.881 | – | 0.769 | – | 0.455 | 1.29 |
| | Group | 2.233 | 0.649 | <0.001 | 1.161 | | |
| | Age | −0.015 | −0.027 | 0.846 | 1.172 | | |
| | Height | 2.745 | 0.124 | 0.385 | 1.241 | | |

Notes.

Abbreviations: ABH, abductor hallucis; FDB, flexor digitorum brevis; VIF, variance inflation factor; Std. Error, standard error.

## DISCUSSION

This study is the first to use ultrasonography to compare IFM and plantar fascia differences between basketball and soccer players. Findings on FHB and ABH thickness confirm sport-specific adaptations due to distinct athletic demands. These results align with the initial hypothesis that morphological differences between groups would be reflected in intrinsic foot muscle thickness. While the present study does not directly assess sport-specific physical demands, the observed differences may reflect underlying adaptive responses to the distinct biomechanical profiles of each sport (*Taylor et al., 2017*; *Domínguez-Díez et al., 2021*; *Song & Deng, 2023*).

There have been limited studies quantifying IFM in soccer and basketball players to date. In this regard, we observed no significant differences in the thickness of the FHB, QP, and PF between players from these different sports. This similarity may be attributed to

shared physical demands in both disciplines, such as frequent accelerations, decelerations, rapid directional changes, and explosive short-distance sprints (*Taylor et al., 2017*; *Tosovic et al., 2012*; *Harper, Carling & Kiely, 2019*). These actions require robust stabilization and force transmission through the foot, likely eliciting comparable neuromuscular activation patterns. Functionally, the FHB plays a key role in metatarsophalangeal joint stabilization and propulsion, the QP contributes to toe flexion and mediates force direction during gait, and the PF maintains arch integrity under dynamic loading. Such biomechanical roles are consistently recruited in both sports, potentially explaining the observed morphological convergence despite differing sport-specific movement profiles.

In opposition to these findings, basketball players possessed significantly thicker ABH ($12.8 \pm 0.42$ mm *vs.* $10.8 \pm 0.39$ mm; $p = 0.002$; ES = 1.15) and FDB ($11.0 \pm 0.17$ mm *vs.* $8.72 \pm 0.40$ mm; $p = 0.001$; ES = 1.89) than soccer players, possibly due to the sport-specific demands they face. In this regard, basketball involves frequent lateral displacements, pivoting maneuvers, and repetitive vertical jumps which impose considerable mechanical loads on the foot and ankle complex (*Chalitsios et al., 2019*). These actions require dynamic stabilization and force transmission through intrinsic foot muscles and the plantar fascia, potentially driving sport-specific morphological adaptations (*Abdelkrim Ben, El Fazaa & El Ati, 2007*). In addition to sport-specific movement patterns, anthropometric and training-related variables may also contribute to the observed morphological differences. In our sample, basketball players were significantly taller, had longer feet, and reported more weekly training hours compared to soccer players. These factors may increase cumulative mechanical load on the intrinsic foot structures, potentially facilitating the development of greater muscle thickness over time.

Supporting this interpretation, previous studies have proposed that the FDB, in conjunction with the PF, plays an important role in attenuating impact forces during repetitive jumping tasks (*Morikawa et al., 2021*). However, in our study, no significant differences in PF thickness were observed between groups. As noted by *Morikawa et al. (2021)*, the PF contributes to shock absorption and medial arch support during landing and rebound tasks; nonetheless, its morphological response may be less sensitive to sport-specific mechanical loading, or may adapt differently than muscle tissue. Accordingly, *Chalitsios et al. (2019)* described a high eccentric force capacity in basketball players, associated with increased jump force ratios, where the first toe and PF act synergistically in propulsion and arch stabilization. Taken together, these findings underscore the functional specificity of each sport and highlight the relevance of tailored preventive and therapeutic strategies (*Chalitsios et al., 2019*).

A key difference between basketball and soccer is the greater frequency of vertical jumps in basketball, where countermovement jump force is closely linked to performance (*Cabarkapa et al., 2024*; *Rodríguez-Rosell et al., 2017*). *Chalitsios et al. (2019)* identified the average rate of force development as a significant predictor of group classification, with higher values in basketball players ($5.67 \pm 2.03$ *vs.* $4.67 \pm 1.65$ kN s$^{-1}$; odss ratio (OR) = 0.30, $p = 0.000$), reflecting their enhanced ability to rapidly generate force during eccentric loading. In contrast, soccer players were characterized by higher propulsive impulse, also a significant predictor (OR = 6.48, $p = 0.047$), with greater values than basketball players

(5.49 ± 0.45 *vs.* 5.25 ± 0.47 N s; $p = 0.020$), suggesting a distinct kinetic strategy focused on impulse generation during the propulsive phase of the movement. These differences in force application strategies may contribute to the sport-specific morphological adaptations observed in our study, particularly the increased thickness of ABH and FDB in basketball players, whose performance relies more heavily on rapid force generation during jumping actions.

Previous studies have underscored the importance of IFM in vertical jumping, not only in their established role of regulating the energetic function of the foot, but also in enhancing ankle joint torque and leverage during the propulsion phase. These findings suggest a potential contribution of IFM to optimizing vertical jump performance in sports characterized by frequent explosive movements, such as basketball. Consistent with our findings, recent research has also examined the relationship between IFM and PF during landing and repetitive rebound jumps—both common actions in basketball—and reported functional differences in FHB and PF across these tasks (*Morikawa et al., 2021*).

## LIMITATIONS AND FUTURE STUDIES

This study presents several limitations that should be acknowledged. Firstly, the sample consisted exclusively of amateur-level male athletes. This limits the generalizability of the findings, as training loads, neuromuscular demands, and structural adaptations may differ significantly in elite or female athletic populations. Secondly, the relatively small sample size reduces the statistical power of the analyses and may hinder the detection of more subtle or nuanced morphological differences. Thirdly, the intrinsic foot muscles were assessed under resting conditions only, without considering their dynamic behavior during functional or sport-specific tasks. This restricts the ability to fully capture the role of IFM under real loading scenarios. Lastly, the cross-sectional design of the study prevents causal interpretations; longitudinal or experimental approaches—particularly those involving targeted exercise interventions—would provide more robust insights into the relationship between IFM morphology and functional adaptation. Future research should include larger and more diverse samples, incorporating both sexes, different competitive levels, and athletes from a broader range of sports disciplines. This would help capture a wider spectrum of mechanical demands and training strategies that may influence IFM morphology. Moreover, integrating ultrasound-derived architectural parameters such as pennation angle and fascicle length, and relating them to performance-based outcomes, could enhance our understanding of sport-specific adaptations and their relevance for injury prevention and rehabilitation.

## PRACTICAL APPLICATIONS

The greater thickness observed in the ABH and FDB muscles among basketball players suggests potential sport-specific morphological adaptations, possibly reflecting the cumulative effect of repeated mechanical loading. Although the present study did not directly evaluate physical demands, it may be hypothesized—based on the functional roles of these muscles and the known movement characteristics of basketball—that these

structures contribute to supporting explosive and multidirectional actions commonly performed in the sport. In turn, the similar thickness of the FHB, QP, and PF between basketball and soccer players indicates that these structures may be subject to comparable functional loading across both sports, likely due to shared requirements for foot stabilization and medial arch support during dynamic actions. Therefore, when designing training or monitoring programs, practitioners may consider focusing on sport-specific loading patterns that influence IFM development, particularly in relation to the muscles that showed distinct morphological differences. These interpretations are further supported by regression models, which identified sport group affiliation as the main factor associated with IFM thickness, emphasizing the potential role of sport-specific mechanical exposure in shaping IFM morphology.

## CONCLUSION

The findings of the present study show an increase in muscle thickness when measured by ultrasound imaging of the FHB and the ABH in basketball players compared with soccer players. These findings suggest sport specific adaptations in intrinsic foot muscles, probably attributable to the different physical demands and training of each sport. While no significant differences were found in the rest of the intrinsic foot muscles, these results highlight the relevance of intrinsic foot muscle strength in both sports and shows the importance of ultrasonography as a measuring tool for muscle morphology. These results were further supported by regression models, which identified belonging to the basketball group—as opposed to the soccer group—as the main factor associated with increased muscle thickness.

### Funding
The authors received no funding for this work.

### Competing Interests
Carlos Romero Morales is an Academic Editor for PeerJ.

### Author Contributions
- Lorena Canosa-Carro performed the experiments, prepared figures and/or tables, and approved the final draft.
- Jorge Hugo Villafañe conceived and designed the experiments, authored or reviewed drafts of the article, and approved the final draft.
- Unai Torres-Berra performed the experiments, prepared figures and/or tables, and approved the final draft.
- Jaime Almazán-Polo conceived and designed the experiments, authored or reviewed drafts of the article, and approved the final draft.
- Helios Pareja-Galeano conceived and designed the experiments, analyzed the data, authored or reviewed drafts of the article, and approved the final draft.

- Sergio Vázquez-González performed the experiments, prepared figures and/or tables, and approved the final draft.
- Carlos Romero-Morales conceived and designed the experiments, analyzed the data, prepared figures and/or tables, authored or reviewed drafts of the article, and approved the final draft.

### Human Ethics

The following information was supplied relating to ethical approvals (i.e., approving body and any reference numbers):

The Research Ethics Committee of European University of Madrid approved this project and the facilities for the study with an internal code CI: 2024-797.

### Data Availability

The raw dataset is available in the Supplementary File.

### Supplemental Information

Supplemental information for this article can be found online at http://dx.doi.org/10.7717/peerj.19773#supplemental-information.

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
