# Peer review of "Ultrasonographic evaluation of intrinsic foot muscle thickness in soccer and basketball players: an observational study"

_PeerJ, doi:10.7717/peerj.19773_

## Round 0.1 · original submission · Major Revisions

Both reviewers have raised some excellent points that should be thoroughly addressed and should strengthen the manuscript further. In particularly, I must add the specific need to justify exploring amateur athletes - for instance, is their participation history and current training/competition demands sufficient to elicit definitive sport-specific variations between the groups. Moreover, the discussion is quite limited, so some further explanation for the lack of variation between groups in some variables and more deeply for the variables that differed between groups.

·

Basic reporting

In general the manuscript is well written and easily followed with a logical flow to the concepts and interpretation.
Title; The authors make reference to football in the title but then in the introduction the reference is to soccer, I suggest being consistent in the sport naming convention and changing the title to ‘soccer’.
Ln 17; Can the authors please reconsider some of the keywords as these should be replicated or use in the manuscript title or ideally in the abstract. The intent is to make the article as easy as possible to find in a database search, so other suggestions to consider include, court and field sports, athletes, muscle morphology, muscle architecture, rehabilitation
Ln 27-38; These should combined into a single paragraph
Ln 27-32; Consider also what impact that the playing surface and the types of footwear worn by athletes participating in each sport has on the IFM adaptation. This aspect to the playing surface and support provided by the footwear is also worth reflection on the interpretation of the findings.
Ln 54; I suggest adding to the end of this sentence “… more readily.” Thus providing further justification for for the use of ultrasound as an imaging technique compared to MRI

Experimental design

The experimental design is suitable for the research question being asked, however check Ln 66 for consistent naming convention.
Ln 71; While the sample size seems to be suitable can the authors provide the details of a power calculation to justify this sample and or was the sample one of convenience?
Ln 75; While I appreciate the description of the criteria, I believe that you need to clarify that the musculoskeletal injuries of concern were only to the lower limb or even just the ankle foot complex?
Ln 89; Can the authors clarify how was a relaxed state or position achieved? The images provided in Figure 1 would suggest that the participant was supporting the weight of the foot on the large toe, would not a soft pad or pillow been more appropriate to support the foot and keep the ankle in a neutral 90 degree position?
Ln 101; Can the authors please confirm what software package was used to perform the USI measurements? In the literature some researchers have used the proprietary collection system software and others have exported the image to then be analysed in software such as ImageJ.
Ln 102; The multivariate predictive model has not been described in the statistical section, please add these details and include justification for which aspect of the research question and hypothesis are being answered.

Validity of the findings

I thank the authors for the clear data set however while all 35 participants are provided there is no indication as to which are sporting cohort each is from. Can a further column be added next to the participant number with 1=Basketball and 2=football?

Additional comments

Ln 144; Can the authors support this opening statement with reference?
Ln 140-147; The authors are discussing here the lack of significant differences in the muscle thickness of the FDB and QP but as the authors collected longitudinal images of these muscles it would be worthwhile if the image quality allowed for the muscle pennation angle to be determined as this would provide greater insight into the rates of shortening versus force production capabilities of the muscles leading to deeper insights into the sporting use adaptations.
Ln 147; Grammatical error, please remove the erroneous closing bracket.

·

Basic reporting

This article is written in English and has correct text. However, there are some uses in which the authors use “soccer” instead of “football” through the Introduction (specifically, lines 27 and 31).

Other grammatical issues include that the in-text citation formatting is inconsistent. For example, line 24 has the citation after the period at the end of the sentence while line 26 has the citation before. I know (and love that) PeerJ allows a free reference format, however it is stated per the guidelines that references should be “Full, clear and consistent” and consistent does not appear to be the case in the Introduction of this manuscript.

I do not believe “USI” is necessary as an abbreviation either, as it is only used 2-3 times. Just spell out ultrasound or ultrasound imaging, please.

Regarding the field background, lines 39-40 state the “recognized importance of the intrinsic foot muscles”. However, the authors have not yet explained this recognized importance. Please add some information in the Introduction regarding this recognized importance prior to this statement. Lines 50-52, “IFM has shown to be a reliable option compared to the gold standard test…” please include a reference at the end of this sentence.

Line 59-60 – “…between sport disciplines Likewise, the study hypothesis…” missing a period after disciplines.

Experimental design

The research question is well defined, and the research is within the Aims & Scope. Within your Methods section, can you please do the following: (1) state how height and body mass were measured, (2) state that shoe size was measured as it was not mentioned until the Results, (3) state what phase of training (e.g., in-season, off-season, pre-season) the athletes were in and how training hours per week was determined (again, not mentioned until Results), (4) define “amateur athletes” (line 72), (5) how were the ultrasound images analyzed- in ImageJ or other software, (6) were any reliability calculations done for the ultrasound examiner and the ultrasound analysis, (7) as I state below regarding the multivariate analysis, you must state that it was done in the Statistical Analysis section of this paper, and (8) provide a rationale on why muscle thickness was used instead of CSA (this should also be mentioned in the Introduction).

Validity of the findings

Overall, the findings are clear. However, there are some cases in which data are repeated in text and in Tables. For example, Table 2 and lines 115-117 are both showing the mean±SD or mean±IR of the muscles by groups. I would prefer to see the mean difference ± the standard error of the mean difference in text and keep Table 2 as is. An example modification would be on lines 114-116, “Significant differences were observed in the ABH and FDB, where basketball players had greater muscle thickness in ABH by 1.96±0.57 mm (95% CI = 0.79 to 3.12 mm) (fill in correct numbers).”

Lastly, for the Results, I believe some of the information can be moved to the Discussion, specifically the end of lines 116-118 and lines 119-123. There is some interpretation of Results here that would be better fit in the Discussion and “beef” up the Discussion more.

The authors have provided the underlying data; however, they have failed to include the most important variable of their manuscript within the data – which groups the Participants belong to! I am assuming, based on me assessing the means and SDs of the data by hand in R, that participants 1-17 are the football group and 18-35 are the basketball group, however this must be included in the data file provided.

Also, the use of the regression model must be defined in the Statistical Analysis portion of the Methods. It is not mentioned until lines 124-131. Further for these models, can Table 3 also include the standard errors of the coefficients along with confidence intervals and the whole model’s p-values? That way, the reader can be given more guidance on model interpretation.

The Discussion section should include more actionable information, similar to what you have in the Clinical Applications. Please add more detail to the Discussion particularly on how the authors believe that these specific IFM muscles could be preferentially trained dependent on the sport (if possible).

Line 146, the “...in both the FHB) and the PF. (20)” delete the ) after FHB.

Line 148-149, “…availability and reliably measuring IFM CSA, comparable to MRI…” one, please use “reliability” instead of “reliably” and two, you utilized muscle thickness, not CSA so this should be stated on why you chose to use thickness in the Introduction.

Line 150-152, “various studies...” what various studies? There are no references here. Please provide some.

Line 153-154, I do not think this belongs here. It is an odd place for this sentence.

Additional comments

Overall, I commend the authors for their work and thank them for allowing me the opportunity to review their manuscript. At its current stage, I believe it requires some major revisions, primarily regarding rationale and clarity on how these findings can be acted upon by practitioners. However, I do not believe my requested revisions are difficult and would gladly review the revised version of this article.

---

## Round 0.2 · Minor Revisions

Thank you for addressing the concerns and suggestions raised by the reviewers for your previous submission - the work has been strengthened further as a result of the changes made. I have thoroughly reviewed the manuscript further and believe some additional changes and considerations are needed to address some aspects before accepting for publication. These suggestions are given below, so please consider them and making any changes where deemed suitable.

#1 Lines 73-74: Please change to "Different sports impose varied physical demands on athletes...".
#2 Line 90: Please end the sentence with an appropriate reference or two after "multidirectional sprinting" here. Then commence the new sentence with "These distinct...".
#3 Line 101: Please cite this sentence given you commence it with "Recent evidence" and provide p-values in reporting results.
#4 Line 106: It seems like a reference should be added at the end of this sentence supporting the the role of IFM function in these movements.
#5 Line 113: Please describe all abbreviations on first use - in this case, MRI. Please check this throughout for other abbreviations such as ICC, SEM, MDC, etc.
#6 Line 124: The term "plantar fascia" has been used throughout the introduction but is instead listed here and abbreviated. Consider abbreviating on first use and then using this abbreviation thereafter as this is not consistent. Also, muscle names are common nouns so do not need to be capitalised.
#7 Line 126: You are not directly measuring the "demands" placed on athletes, so just remove "and demands" on this line please.
#8 Lines 126-128: You don't explicitly state a hypothesis with reasoning for it. In this case, if you hypothesise differences, what differences did you think would be identified and why (with citations) regarding muscle thickness between the two groups? Please improve the clarity of the hypothesis statement made here.
#9 Line 142: Why was a "large" effect chosen here in the power analyses? Was this based on other similar research regarding typical effects, was this the magnitude of difference you wanted to detect, was another reason applied? I see you mention further down that the large effect supports the feasibility of the final sample able to be recruited, but the sample still falls short of the required number of participants to detect such an effect. Perhaps justify the choice of the effect size here.
#10 Line 153: No need to capitalise "No" here.
#11 Lines 155-157: Perhaps you could just state "Players were excluded if they failed to meet all eligibility criteria and possessed a history of lower-limb injuries, surgeries, or conditions that could impact muscle morphology" for the exclusion statement. You seem to list off many conditions but it seems like you were strictly concerned with any that could impact muscle morphology?
#12 Line 165: Change "weight" to "body mass" here and throughout given "weight" is a force not a mass.
#13 Lines 176-180: Please change this sentence for clarity to: "High within-session intra-rater ICC >0.94 (SEM <3.6%; MDC <10.0%) alongside reasonable between-session intra-rater ICC >0.81 (SEMs <6.5%; MDCs <17.9%) have been reported [18]." I am assuming reference 18 contains these data, but if not, please adjust and cite appropriately. Also, there is no need to mention inter-rater reliability given only one rater performed all ultrasound assessments in your study.
#14 Line 191: Here you use "subjects", so please be consistent with referring to the athletes you tested as "participants" when referring to them. Please check this usage throughout.
#15 Line 195: You use "structures" twice within a few words here, can you use a different term for one of these?
#16 Lines 211-212: Can you possibly give ranges for each effect size magnitude descriptor. For instance, was small equal to or less than 0.2? Was medium from 0.21 to 0.50? And was large equal to or greater than 0.8? Ranges work better than a single cutpoint given to describe effect magnitudes for clarity.
#17 Line 234: Please be consistent with abbreviations as you abbreviate the different points of measurement earlier but list them in full here. Please check throughout for consistency in abbreviation use.
#18 Lines 236-241: It seems like the first three sentences in this paragraph are describing (or repeating) the statistical analyses. Can you move any necessary detail to the statistical analyses section from here please and just keep to reporting the results from the regression analyses here?
#19 Lines 244-245: Please remove "the model" before "also reached" here.
#20 Line 250: Remove "To the author's knowledge" here and just begin with "This study is the..." here.
#21 Lines 254-255: Your work does not support that players from soccer and basketball face different demands, your research only supports that the muscle thickness in their feet differ. Yes, this may be speculated to be as a result of varied adaptive processes dependent on the demands they face, but this is only a mechanistic explanation from your findings, not something that is directly supported by your data. In this case, please adjust this sentence and statements made elsewhere of this nature for greater accuracy.
#22 Lines 257-261: This paragraph offers very little insight, so I would suggest adjusting it to: "There has been limited studies quantifying IFM in soccer (CITE ANY STUDIES) and basketball (CITE ANY STUDIES) players to date. In this regard, we observed no significant differences in FHB, QP, and PF between players from these different sports." Then explaining what physical demands (or other aspects) may be similar to promote comparable muscle morphology for these muscles specifically between the two sports with citations to support the reasoning provided. This reasoning should relate to the specific functions and actions of the muscles listed here to show how the mechanical loading between the two sports might promote them to be similar.
#23 Lines 262-267: Please adjust these first two sentences for simplicity here to: "In opposition to these findings, basketball players possessed significantly thicker ABH (12.8 ± 0.42 mm vs. 10.8 ± 0.39 mm; p = 0.002; ES = 1.15) and FDB (11.0 ± 0.17 mm vs. 8.72 ± 0.40 mm; p = 0.001; ES = 1.89) than soccer players, possibly due to the sport-specific demands they face. In this regard, basketball involves...".
#24 Line 268: Please cite work showing basketball involves these demands at the end of this sentence after "jumps".
#25 Line 273: Here you mention the PF is also involved, but this was not different between groups in your study? Does this reasoning contradict your findings then here?
#26 Line 275: You use the term "athletes" here, but use "players" mostly elsewhere - please be consistent with whatever term you adopt throughout.
#27 Line 274: Please cite the reference number instead of "(2019)" here as per journal requirements and you have done elsewhere. Check this elsewhere such as on line 281.
#28 Line 275: Here you say "greater eccentric force generation capacity in basketball athletes", so is this from a direct comparison between soccer and basketball players? If so, please state this more clearly.
#29 Line 287: Please include a final sentence for this paragraph bringing this reasoning back to your results. In other words, how do these variations in kinetic strategies translate to underpinning your findings? Perhaps this is somewhat done in the next paragraph, so you could effectively link them together here.
#30 Line 301: Your focus was on soccer and basketball players, so why are wider disciplines needed here? What will expanding to other sports provide precisely? Perhaps you could justify this recommendation with an extra sentence here for clarity.
#31 Limitations: Consider strictly mentioning the recruitment of amateur-level athletes as a limitation given the structure and nature of the training/physical stress encountered will greatly differ between competition levels - with perhaps more impact being given with investigation of higher-level athletes. Also, you need to mention the small sample size as a limitation (given the study is underpowered) and the focus only on males. There are other limitations you could mention, but I feel this paragraph needs some greater depth with clear mention of limitations and then future research that should be done to extend on your study further.
#32 Practical applications: This paragraph is not really clear in terms of translation of your results. For instance, please clearly state the greater thickness in the ABH and FDB muscles in basketball players suggests.... In turn, the similar thickness of other muscles between basketball and soccer players indicates... At present you state soccer players should prioritize training to enhance foot stability and agility, which can be true for basketball (among other sports), so it doesn't really stem from your findings directly. The statement about reducing injury risk and optimizing performance is also not directly an implication of your results given you did not examine these aspects, so again should be removed or modified for better relation to your actual findings. Please adjust this section accordingly.
#33 Consideration: The basketball players were taller with larger feet, plus completed more weekly training hours (Table 1). These factors seem like they could help explain some of the findings perhaps? You could mention them more directly as mechanisms for your findings within your discussion when providing reasoning if applicable.

·

Basic reporting

No comment

Experimental design

No comment

Validity of the findings

No comment

·

Basic reporting

The authors have fulfilled my recommended edits regarding the basic reporting of this manuscript.

Experimental design

No comment. The authors have done a great job revising their manuscript.

Validity of the findings

No comment. The authors have improved the manuscript significantly.

Additional comments

No comment.

---

## Round 0.3 · accepted · Accept

Thank you for addressing all of the revisions suggested for the previous version of this manuscript. The submission has been greatly strengthened and is suitable for acceptance.